# Evaluation of Abiotic Controls on Windthrow Disturbance Using a Generalized Additive Model: A Case Study of the Tatra National Park, Slovakia

**Vladimír Falťan** [1], **Stanislav Katina** [2], **Jozef Minár** [1], **Norbert Polčák** [3], **Martin Bánovský** [4], **Martin Maretta** [5], **Stanislav Zámečník** [2] and **František Petrovič** [6,*]

1   Department of Physical Geography and Geoecology, Faculty of Natural Sciences, Comenius University in Bratislava, Mlynská dolina, Ilkovičova 6, 84215 Bratislava, Slovakia; vladimir.faltan@uniba.sk (V.F.); jozef.minar@uniba.sk (J.M.)
2   Institute of Mathematics and Statistics, Faculty of Science, Masaryk University, Kotlářská 267/2, 61137 Brno, Czech Republic; katina@math.muni.cz (S.K.); zamecnik@math.muni.cz (S.Z.)
3   Department of Meteorological Forecasts and Warnings, Slovak Hydrometeorological Institute, Jeséniova 17, 83315 Bratislava, Slovakia; norbert.polcak@shmu.sk
4   T-MAPY Slovensko Ltd., Medený Hámor 15, 974 01 Banská Bystrica, Slovakia; martin.banovsky@tmapy.sk
5   Esprit Ltd., Pletiarska 2, 96901 Banská Štiavnica, Slovakia; maretta@esprit-bs.sk
6   Department of Ecology and Environmentalistics, Faculty of Natural Sciences, Constantine the Philosopher University in Nitra, 94901 Nitra, Slovakia
*   Correspondence: fpetrovic@ukf.sk; Tel.: +421-907-756-489

**Abstract:** Windthrows are the most important type of disturbance occurring in the forests of Central Europe. On 19 November 2004, the strong northeastern katabatic winds caused significant damage and land cover change to more than 126 km$^2$ of spruce forests in the Tatra National Park. The risk of subsequent soil erosion and accelerated runoff has increased in the affected habitats. Similar situations may reoccur this century as a consequence of climate change. A geographical approach and detailed research of the damaged area with more comprehensive statistical analyses of 47 independent variables will help us to obtain a deeper insight into the problem of windthrow disturbances. The results are based on a detailed investigation of the damaged stands, soil, and topography. A comprehensive input dataset enabled the evaluation of abiotic controls on windthrow disturbance through the use of a generalized additive model (GAM). The GAM revealed causal linear and nonlinear relationships between the local dependent quantitative variables (the damage index and the uprooting index) and independent variables (various soil and topography properties). Our model explains 69% of the deviance of the total damage. The distribution of the wind force depended upon the topographical position—mainly on the distance from the slope's foot lines. The soil properties (mainly the soil skeleton, i.e., rock fragments in stony soils) affect the rate and manner of damage (uprooting), especially on sites with less wind force. Stem breakage with no relation to the soil prevailed in places with high force winds. The largest number of uprooted trees was recorded in localities without a soil skeleton. The spruce's waterlogged shallow root system is significantly prone to uprooting. The comprehensive research found a significant relationship between the abiotic variables and two different measures of forest damage, and can expand the knowledge on wind impact in Central European forests.

**Keywords:** spruce forests; wind disturbances; climate change; abiotic factors; generalized additive model; biodiversity conservation; forest ecosystem management; Slovakia

## 1. Introduction

There is increasing interest in how large, infrequent, natural disturbances affect both vegetation [1] and human society. With reference to the general definition provided by Pickett and White [2], the disturbance of a forest ecosystem happens discretely over time, and disrupts the ecosystem's structure, composition, and processes by altering its physical environment and resources, causing the destruction of plant biomass. Strong winds pose a number of hazards in different areas, such as structural safety, aviation and wind energy [3]. Windthrow is the most important type of disturbance in the forests of Central Europe [4]. Moreover, wind events may reoccur more frequently in the current century due to climate change, which has the potential to invalidate historical baselines by altering the key drivers of disturbance regimes [5]. The adaptation to new conditions requires a deeper understanding of all of the controlling factors of wind disturbance. Stand composition is assumed to have an influence on the vulnerability of forests to storm damage [6]. Although the leading role of atmospheric and vegetative factors is indisputable, other factors may also play an important role—largely on a local scale. The biotic factors that influence the extent of the damage mainly include the tree species, the stem size (e.g., trunk diameter, tree height, and stem/crown weight), and the presence of pathogens [7]. The most frequently considered abiotic factors include the wind velocity, the direction topography, and the soil characteristics (e.g., [7,8]).

Quantitative models are powerful tools in analyzing the complex relations between disturbances and their environment, as well as their interactions with forest management [9]. Regarding wind damage research, there are two basic modeling approaches: (1) statistical/empirical modeling and (2) mechanistic modeling. The investigation of the dependence of natural disturbances on stand and site conditions is most frequently based on regression analyses. For the modeling of storm damage in spruce forests, Lohmander and Helles [10] used a logistic regression model (RM), and used the soil conditions, tree height, and tree age as predictors (independent variables); Vallinger and Fridman [11] used a logistic RM, and used the altitude, topographical position, stand age, and site quality as predictors; Mikita and al. [12] used a discriminant analysis, and used the elevation, exposure, edaphic category, distance from forest edges, stand density, and stand height as predictors. The logistic RM was also used for the investigation of biotic and abiotic variables in other types of forest stands from North America, e.g., [8,13,14]. Kramer et al. [8] used exposure, slope, elevation, and soil texture as predictors in a multiple RM. Wang and Xu [15] developed a generalized linear RM; Albrecht et al. [16] used generalized linear mixed models and boosted regression trees for storm damage risk in southwest Germany. Hanewinkel et al. [17] and [18] used neural networks for the assessment of the risk of windthrow. Kenderes et al. [19] investigated the role of the topography and tree stand characteristics through the use of non-parametric statistical tests, and through a series of classification and regression tree analyses.

Klaus et al. [20] performed a logistic RM to create a storm damage sensitivity index for North Rhine-Westphalia, based on damage data from the 'Kyrill' storm. There are also more studies which are based on a broad forestry database [6,8,13,14,21–23]. The advantage of an official forest inventory is the relatively large number of records, and the monitoring of the forest dynamics. A potential disadvantage of recording forest samples in a regular grid is that it may not capture completely all the types of ecosystems in areas affected by disturbances. The dependent variable was often estimated from aerial images, mostly on a binary scale (damaged and non-damaged areas) or an ordinary scale; the independent variables included a mix of quantitative and qualitative data. However, a deeper insight into the problem requires the use of more precise input data, as more comprehensive statistical analyses capture both linear and non-linear dependencies. Detailed field research can provide a valuable database for the modeling of an ecosystem [24–32]. The site conditions (including soil, topography) and climate are also used in biogeographical research linking the functional biogeography of ecosystems [33].

The previous research has mostly focused on the analytical investigation of the chosen biotic or abiotic factors. In order to bring qualitatively new complex results, we focused on the combination

of several methodological approaches, with the task of capturing the influence of topographic and edaphic factors on the degree of the storm damage to forest vegetation more comprehensively in our research.

The purpose of our study is to investigate abiotic controls on the windstorm impact in the selected representative localities of the High Tatras foothills using a generalized additive model (GAM), and to develop models for both (1) the overall tree damage caused by wind and (2) the uprooting damage based on our detailed empirical database, which was collected using an integrative approach.

## 2. Materials and Methods

### 2.1. Wind Event and Study Area

The Tatra National Park is the oldest national park in Slovakia. It was established to protect Carpathian high mountain biotopes and their biodiversity. Together with the Polish Tatra National Park, the Tatras have been a UNESCO Biosphere Reserve since 1993. Spruce monocultures were planted in the outskirts of the national park in the last century.

Windstorms periodically affect the ecosystems of spruce forests in the foothills of the Tatra Mts. (Figure 1). The local winds and intensive, but rare, boras are conditioned by the orography of the highest mountains of the Carpathians. The Western Carpathians are part of the Alpine–Himalayan orogenic belt and the northernmost branch of the alpine orogeny [34].

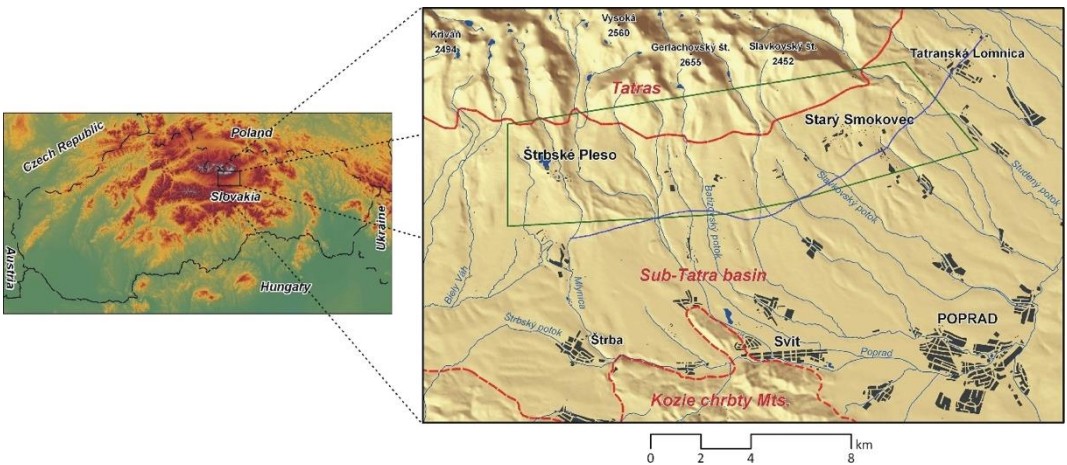

**Figure 1.** Situation map of the case study area. Green line: the study area; full red line: the boundary of geomorphic units (primary foot line); dashed red line: the boundary of the geomorphic sub-units; blue line: the front of the glacial moraines (secondary foot line).

The calamity, which occurred on 19 November 2004, may be considered exceptional for its spatial extent. A frontal border was formed over the inland part of Europe, dividing the warmer air over the southern part of the continent from the colder air over the northern part of Europe. Sea air, originally of arctic origins, started to flow along its western part as a result of the high-pressure gradient (Figure 2). Within the area containing the ridges of the High and Low Tatras, maximum wind gusts were recorded by the Slovak Hydrometeorological Institute as follows: Lomnický Štít peak (166 km/h), the upper leeward mountain slope areas in Skalnaté pleso tarn (194 km/h), and on the foothills of Stará Lesná (162 km/h). More than 126 km$^2$ of forests were damaged, with a wood volume of 2.3 million m$^3$ located at altitudes between 700 and 1350 m. The wind caused much damage, and it even contributed to the death of one person. 84,000 cubic meters of timber had to be removed from the intra-urban areas of various Tatra settlements. The total amount of damage estimated by the European Commission was close to 195 million euros. The most affected areas were on the foothills. In recent years, the risk of flash floods due to climate change and deforestation has increased. For example, in the sub-Tatra

counties of Poprad and Kežmarok, they dealt with an emergency situation due to rains and local floods in July 2018.

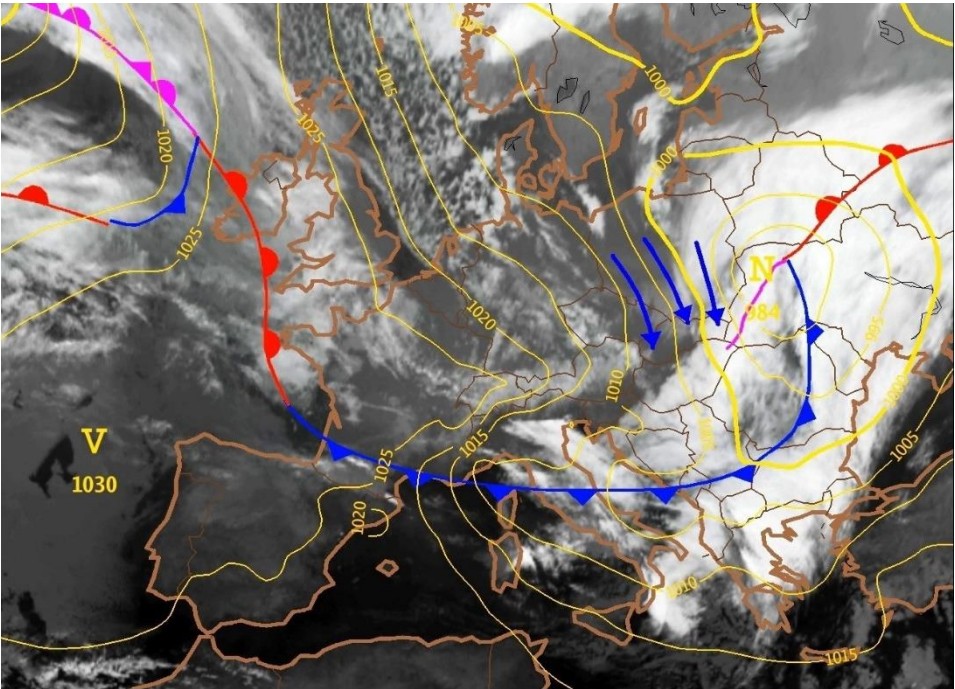

**Figure 2.** Synoptic chart on 19 November 2004, 18:00 UTC (Slovak Hydrometeorological Institute).

The windthrow changed the land cover, especially in the High Tatras foothills [35], damaged a variety of habitats with differing intensities, and also affected the diversity of vertebrate and invertebrate communities [36,37]. Part of the windthrow area was left unmanaged for natural development, but most of the area was cleared. The sub-mountain and mountain floodplains are covered by fragments of communities of *Alnenion glutinoso-incanae* (Oberd. 1953) and *Eu-vaccinio-Piceenion* (Oberd. 1957). Coniferous forests represented most of the Tatras' vegetation. In higher areas of the National Natural Reservations, the level of the vegetation damage is lesser, with natural generic structures dominating, especially communities of *Eu-Vaccinnio-Abietion* (Oberd. 1957), and the local presence of *Vaccinio-Abietenion* (Oberd. 1962) and *Piceion excelsae* (Pawlowski in Pawlowski et al. 1928). Cambic podzols dominated in the study area. Spruce forests in the sites of glacifluvial and polygenetic forms with waterlogged soils were among the most damaged areas. The vegetation in glacial moraines was less damaged. The vegetation of the fluvial land systems with Fluvisols belong to relatively less damaged areas. The least damaged forests are in the sites of fault lee slopes on granodiorite and polygenetic forms on glacial sediments [35]. The area belongs to the region with a mean annual temperature of 5.3 °C and a mean annual precipitation of 800–900 mm. Fluvioglacial sediments and podzolic cambisols predominate in the area. The soil granularity is mostly clay to sand clay [38]. Permanent research areas, which represent both undisturbed and extracted windthrow and reference forest, were established in the region by the administration of the national park for long-term research [39].

The wind-disturbed stands of the Tatras are also responsive to bark-beetle invasions in the long term. Furthermore, an outbreak of *Ips typographus* was recorded before this event, from 1990 to 2000 [40]. As a result of regular disturbance, a specific community of spruce forests has formed in the Tatra region at altitudes of up to 1200 m a.s.l., known as larch–spruce forests (*Lariceto-Piceetum*) and pine-spruce forests (*Pineto-Piceetum*). The species composition and quantity of the natural regeneration is reflected by the influence of both windthrow and tree extraction [41].

## 2.2. Variables

This particular wind event initiated extensive research regarding its consequences and causes, e.g., [35,36,38,39,41–44]. Two main directions of wind flow were observed by the station network of the Slovak Hydrometeorological Institute: from the north (N) and northwest (NW). The NW direction predominated. The evaluation of the orientation of the georelief (EG) belongs to crucial parts of windthrow disturbance investigation. The exposure to N and NW winds (EGNW, *EGNWW*) was defined as the difference of the slope aspect and N respective to the NW direction. Because the wind came from the 30° mountain slope, the "3D exposition to N/30° wind" was also computed as a space angle of the incidence of the predominant wind flow (WIA).

We focused further on the influence of the site's condition on the forest damage (Figure 3). An integrative approach [45] was used for the field investigation of the local disturbances and site conditions on the 260 sample plots (research areas with sizes of approximately 100 m$^2$, characterized by soil pits) during the 2006–2008 summer seasons. The plots were localized as neighboring more or less-damaged local ecosystems for the recording of the local differences between them. The location of the plots was focused using a global positioning system GPS. We described the local characteristics of the vegetation (including the numbers of standing, uprooted, and broken trees), wind direction and topography, and hydrological and soil conditions.

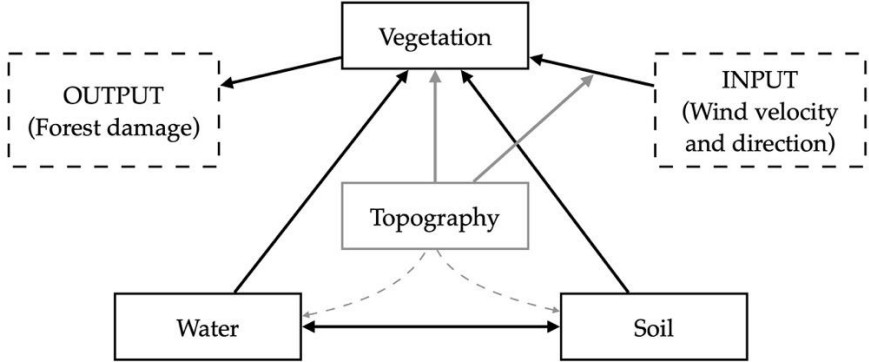

**Figure 3.** System scheme of influence of site conditions on the windthrow event.

The geographical positions of the sample plots were determined using a GPS receiver and interpreted in the ArcInfo 9.3 GIS environment (ESRI).

The variables that firstly reflect forest damage, and secondly could influence the local wind velocity and tree stability (the main character of the root system) were used in the analysis (see Abbreviations). The majority (primary variables) were determined directly in the field by a simple count (the numbers of damaged and standing trees, and trees of a particular species); measurement by compass, measuring tape and ruler (topography and soil horizons thickness, size of soil skeleton (rock fragments), depth of water table); and estimation (abundance of tree layer, soil texture, volume of rock skeleton in the soil, and geometric type of landform). The rock skeleton characteristics (weathering, maximum and average size) were derived from the measurement of some tens of rock fragments in a given soil pit. Based on our previous experience [42,43,46], the secondary variables were subsequently derived from the primary variables, e.g., exposure to the wind and index of dryness, etc. (see Abbreviations). Specific attention was paid to the distance from the ridges and the slope foot lines (in both cases, the main and secondary lines were distinguished, as both could influence the local distribution of wind power). The foot lines represent boundaries between differently inclined areas, usually at the contact of a hillside and a valley. The primary foot line follows the tectonic boundary between the Podtatranská kotlina (basin) and the Tatras. The secondary foot line follows the front of the Tatra's glacial moraines. The forest damage was reflected by the number of damaged trees (uprootings and stem breakages) and the number of standing trees. However, all of these measures did not reflect the

initial state: the density of the unbroken forest stands. Therefore, a damage index *Di* [47] (the ratio of damaged trees to all of the standing trees before the disturbance; from 0 (all trees are standing) to 1 (totally damaged)) and an uprooting index *Upi* (the ratio of uprootings to all of the standing trees before the disturbance; from 0 (all trees are standing) to 1 (totally damaged)) were used to define the relative forest damage (see Abbreviations).

Stating an exact representation of a particular tree species before the wind event was problematic. The research was realized after the disaster, and part of the salvaged timber had been extracted. Only the composition and maximum height of the standing trees was systematically recorded. We also studied the abundance of the tree species which prevailed (rPicea, rPinus, rLarix, rBetula). Abundance was relevant also for the linear-circular correlation analyses (Table 1).

**Table 1.** Results of the linear-circular correlation analyses. *n*—sample size; *r*—correlation coefficients; LB—lower bound and UB—upper bound of 95% confidence intervals for *r*; *z*-statistic—Fisher *z*-statistic and *p*-value; *Upi*—uprooting index; *Di*—damage index; EG—the orientation of the georelief to the north; EGNW—the orientation of the georelief equal to the direction of the N wind; *EGNWW*—the orientation of the georelief equal to the direction of the NW wind; WIA—the angle of impact of the wind. The significance level was Bonferroni corrected by dividing 0.05 by 4 (the table is divided into seven sections by four correlations), which resulted in 0.0125. All of the *p*-values smaller than this level indicate statistically significant correlations at a Bonferroni corrected significance level of 0.0125.

| Variable | *n* | *r* | LB | UB | *z*-Statistic | *p*-Value |
|---|---|---|---|---|---|---|
| *Upi* vs. EG | 245 | 0.247 | 0.1261 | 0.3616 | 3.9318 | 0.00008 |
| *Upi* vs. EGNW | 245 | 0.176 | 0.0522 | 0.2952 | 2.7727 | 0.00556 |
| *Upi* vs. *EGNWW* | 245 | 0.267 | 0.1469 | 0.3799 | 4.2625 | 0.00002 |
| *Upi* vs. WIA | 250 | 0.138 | 0.0137 | 0.2572 | 2.1757 | 0.02958 |
| *Di* vs. EG | 242 | 0.167 | 0.0419 | 0.2872 | 2.6083 | 0.00910 |
| *Di* vs. EGNW | 242 | 0.127 | 0.0005 | 0.2487 | 1.9679 | 0.04908 |
| *Di* vs. *EGNWW* | 242 | 0.187 | 0.0625 | 0.3060 | 2.9272 | 0.00342 |
| *Di* vs. WIA | 247 | 0.128 | 0.0035 | 0.2491 | 2.0142 | 0.04399 |
| *Upi* vs. EG, EG 0–90 | 68 | 0.089 | −0.1523 | 0.3209 | 0.7223 | 0.47013 |
| *Upi* vs. EG, EG 90–180 | 148 | 0.140 | −0.0220 | 0.2945 | 1.6944 | 0.09018 |
| *Upi* vs. EG, EG 180–270 | 25 | 0.181 | −0.2306 | 0.5377 | 0.8588 | 0.39047 |
| *Upi* vs. EG, EG 270–359 | 4 | 0.995 | 0.7661 | 0.9999 | 2.9708 | 0.00297 |
| *Di* vs. EG, EG 0–90 | 68 | 0.170 | −0.0715 | 0.3923 | 1.3824 | 0.16686 |
| *Di* vs. EG, EG 90–180 | 146 | 0.128 | −0.0357 | 0.2841 | 1.5332 | 0.12522 |
| *Di* vs. EG, EG 180–270 | 24 | 0.149 | −0.2709 | 0.5209 | 0.6867 | 0.49230 |
| *Di* vs. EG, EG 270–359 | 4 | 0.999 | 0.9385 | 1.0000 | 3.6853 | 0.00023 |
| rPicea > 0, *Upi* vs. EG | 101 | 0.262 | 0.0707 | 0.4356 | 2.6608 | 0.00780 |
| rPicea > 0, *Upi* vs. EGNW | 101 | 0.250 | 0.0572 | 0.4245 | 2.5266 | 0.01152 |
| rPicea > 0, *Upi* vs. *EGNWW* | 101 | 0.247 | 0.0540 | 0.4219 | 2.4953 | 0.01258 |
| rPicea > 0, *Upi* vs. WIA | 102 | 0.220 | 0.0269 | 0.3977 | 2.2277 | 0.02590 |
| rPicea > 0, *Di* vs. EG | 99 | 0.229 | 0.0336 | 0.4084 | 2.2891 | 0.02207 |
| rPicea > 0, *Di* vs. EGNW | 99 | 0.149 | −0.0528 | 0.3339 | 1.4418 | 0.14935 |
| rPicea > 0, *Di* vs. *EGNWW* | 99 | 0.229 | 0.0327 | 0.4076 | 2.2801 | 0.02260 |
| rPicea > 0, *Di* vs. WIA | 100 | 0.236 | 0.0412 | 0.4130 | 2.3660 | 0.01798 |
| rLarix ≥ 0.4835 *Di* vs. EG | 25 | 0.523 | 0.1617 | 0.7611 | 2.7253 | 0.00642 |
| rLarix ≥ 0.4835 *Di* vs. EGNW | 25 | 0.523 | 0.1617 | 0.7611 | 2.7253 | 0.00642 |
| rLarix ≥ 0.4835 *Di* vs. *EGNWW* | 25 | 0.367 | −0.0327 | 0.6657 | 1.8065 | 0.07085 |
| rLarix ≥ 0.4835 *Di* vs. WIA | 25 | 0.540 | 0.1848 | 0.7710 | 2.8369 | 0.00456 |

The description of all of the other variables used in the paper is in the Abbreviations section.

## *2.3. Models*

The statistical analyses of the abiotic controls on windthrow were performed in the R computing environment, R version 3.6.3 [47], using integrated development environment RStudio Desktop Version 1.2.5042 (RStudio PBC, Boston, MA, USA).

The association of *Upi* and *Di* with the variables related to the orientation (exposition) of the georelief variables (EG, EGNW, *EGNWW*, WIA) was assessed by the use of a linear-circular correlation coefficient [48]. In this situation, a classical (linear-linear) correlation coefficient couldn't be used,

since the variables related to the orientation are directional variables. The null hypothesis, that the correlation coefficient is equal to zero against a two-sided alternative, was tested by the one-sample Fisher *z*-test [49]. All of the alternative hypotheses were two-sided, and the statistical tests were performed at a significance level equal to 0.05. The correlation analyses were carried out for the orientation of the georelief variables in the whole sample, and were also separated into quadrants. In order to identify the best thresholds for the relative abundance of the prevailing tree species, i.e., the optimal values for the separation of the class labels, with respect to *Upi* and *Di*, we used regression trees [50]. The regression models were defined as *Upi = rPicea + rLarix + rPinus + rBetula* and *Di = rPicea + rLarix + rPinus + rBetula*. Then, the linear-circular correlation coefficient was calculated in subgroups with relative abundances below and above these thresholds. The statistically most important thresholds are—0 for rPicea with respect to both *Upi* and *Di*, and 0.4835 for rLarix with respect to *Di*.

GAM is a generalized linear model in which the dependent variable depends linearly on the unknown smooth functions of some independent variables, and interest is focused on statistical inferences about these smooth functions, which here are penalized cubic regression splines. GAM does not assume a priori any specific form of this relationship (compared to the ordinary linear regression model (OLRM), with the relationship defined as a line) and can be used to reveal and estimate the non-linear effects of the independent variable on the dependent variable. Using GAMs allows for more flexible modelling, where the degree of smoothness can be estimated as part of the model fitting using generalized cross validation.

We submitted data to GAMs built up on penalized regression splines [51] as part of the *mgcv* 1.8-33 R package in order to determine any causal relationships between the dependent variables (damage index (*Di*) and uprooting index (*Upi*)) and a set of independent variables (Table 1). In OLRMs, the parameters are estimated by the minimization of ordinary least squares (OLS). These types of models might not be appropriate in our case due to the possible nonlinear relationships between the dependent variables and the set of independent variables. In this situation, the GAM is more appropriate. The methodology behind the GAM has greater flexibility than the traditional ordinary linear regression model (OLRM). It relaxes the usual linear parametric assumptions, and enables researchers to uncover nonlinear structures in the relationships between the independent variables and the dependent variables that might otherwise be overlooked. Here, OLS is replaced by penalized least squares [51]. Each of the individual additive terms (all smooth functions of covariates (independent variables)) is estimated using a univariate smoother, termed as a penalized regression spline (here, we used a penalized cubic regression spline). The smoothing parameters were chosen via a generalized cross validation in order to maximize the ability of the model to predict the data to which it was not fitted [51]. This algorithm is applied to each variable (as an additive term) separately, and all estimated parameters together form the GAM. For each dependent variable, the sub-optimal model was selected during the model selection process from the saturated model (which included all of the variables mentioned in the Abbreviation section) using Akaike information criterion, applying double penalty approach, accounting for null space as in [52], Section 2.1. The null hypotheses for the smooth terms were defined such that it is not a (potentially non-linear) association of a dependent variable with a covariate, given (potentially non-linear) association with other covariates. The null hypotheses were tested against two-sided alternatives. There were also some covariates with higher related *p*-values in the model. We consider *p*-values between 0.05 and 0.1 as being marginally statistically significant, and other higher *p*-values in the model as being statistically non-significant but having valuable contributions to the model's quality.

## 3. Results

### 3.1. Results of the Linear-Circular Correlation and Statistical Models

Based on our field records, the most abundant standing tree was *Picea abies* (66.07%), but there were three other tree species with an abundance of more than 5%—*Pinus silvestris* (6.40%), *Larix decidua*

(13.94%), and *Betula pendula* (5.63%). From these biotic characteristics, the maximum height of the standing trees was the most important factor influencing the forest damage [10,35,36]. The uprootings also affected the larch tree, but to a lesser extent, due to its stronger lateral root system. Stem breakages of *Picea abies* and *Larix decidua* were also located on different exposed slopes. Several other associations have a nonlinear character (Table 1).

The investigation of the relationship between the abiotic characteristics and the degree of wind damage of the tree layer on 260 tesserae using GAM yielded the following (most relevant; others not shown) results (*f* is a smooth function—a penalized regression spline—for a particular nonlinear relationship; the other relationships are linear, and one is quadratic):

$$Di = H1Sx + ID + Alt + DMF + f_1(GSa) + f_2(H2Sv) + f_3(G) + f_4(DSF),$$

where *Di* means the damage index, *H1Sx* is the maximum topsoil skeleton size, *ID* is the index of dryness, *Alt* is the altitude, *DMF* is the distance from the main slope foot line, *GSa* is the vertical gradient of the skeleton's average size, *H2Sv* is the subsoil skeleton volume, *G* is the slope gradient, and *DSF* is the the distance from the secondary slope's foot line:

$$Upi = H1Tn + ID + f_1(DSF) + f_2(EGNWW) + f_3(G) + f_4(H1Sx),$$

where *Upi* means the uprooting index, *H1Tn* is the topsoil thickness, *ID* is the index of dryness, *DSF* is the distance from the secondary slope's foot line, *EGNWW* is the exposition to the NW wind, *G* is the slope gradient, and *H1Sx* is the maximum topsoil skeleton size.

## 3.2. Abiotic Controls on Wind Disturbance

The model for *Di* characterizes the dependence of the forest damage on the site conditions and topography (Table 2, Figure 4). When *H1Sx* decreases by 10 cm, the *Di* increases by 2.4%; when the *ID* decreases by 1, the *Di* increases by 3.1%; when the Sa increases by 10 cm, the *Di* increases by 3.0% (see *β*s in Table 2).

**Table 2.** Results of the sub-optimal GAM for relative forest damage, linear part. *β*—estimate of the regression coefficient; SD(*β*)—estimate of standard deviation of *β*; *H1Sx*—maximum topsoil skeleton size; *ID*—index of dryness; *Alt*—altitude; *DMF*—distance from main slope foot line; Sa—average size of the soil skeleton.

| Variable | *β* | SD(*β*) | *t*-Statistic | *p*-Value |
|----------|-----|---------|---------------|-----------|
| intercept | −1.1494 | 1.05259 | −1.0993 | 0.279 |
| *H1Sx* | −0.0019 | 0.00074 | −2.6183 | 0.011 |
| *ID* | −0.0349 | 0.01624 | −2.1465 | 0.036 |
| *Alt* | 0.0017 | 0.00093 | 1.8173 | 0.074 |
| *DMF* | 0.0001 | 0.00007 | 1.2912 | 0.201 |

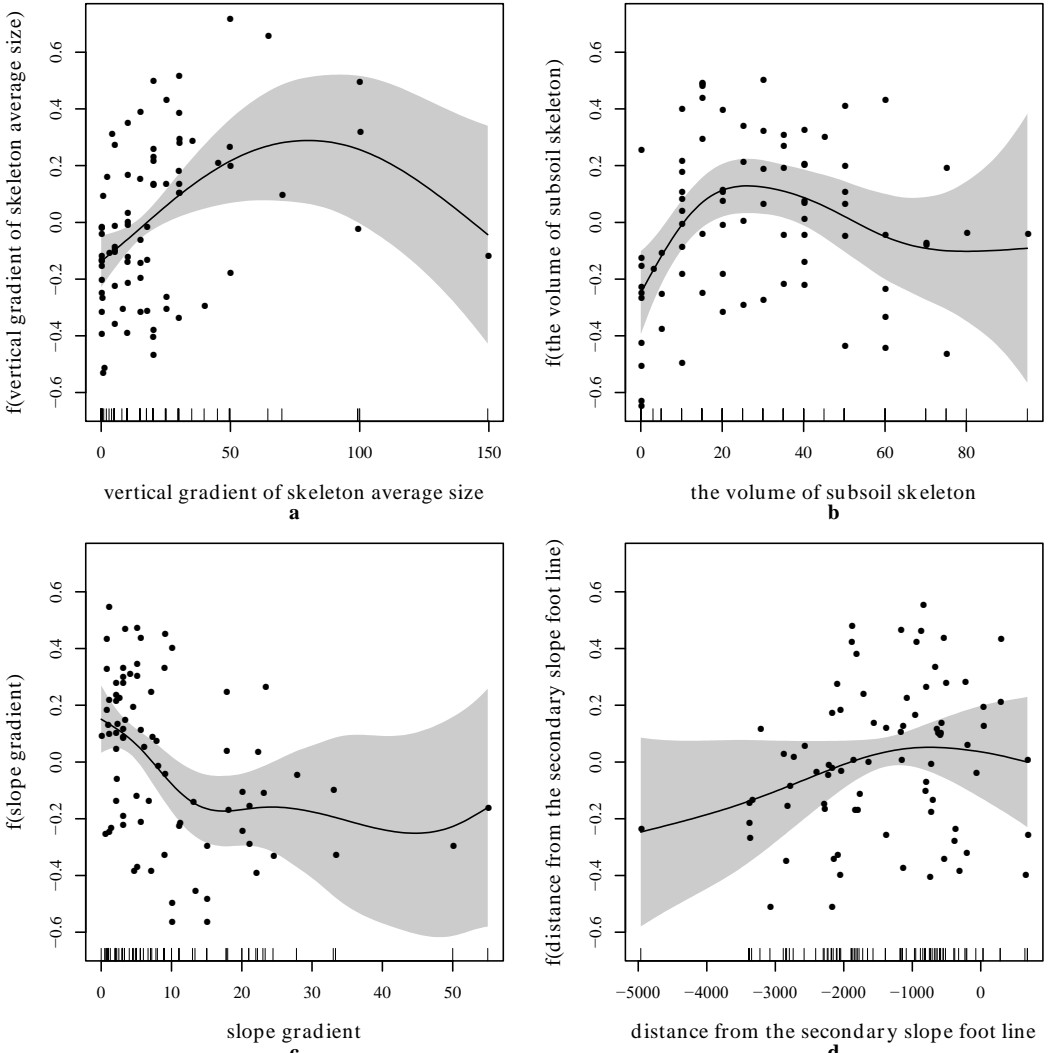

**Figure 4.** Estimated causal relationship of the relative forest damage on the particular variables (spline part of the model for *Di*)—(**a**) vertical gradient of skeleton average size, (**b**) the volume of subsoil skeleton, (**c**) slope gradient, (**d**) distance from the secondary slope line.

The deviance explained by the model's *Di* is 69% (equivalent to $R^2$, not shown). The soil characteristics evidently have the highest importance in this model. The maximum size of the topsoil skeleton (*H1Sx*) has the greatest linear relationship to *Di* (Table 2). The explanation is simple: bigger stones are able to better fixate the tree roots, and especially prevent uprooting.

The model for *Upi* (deviance explained = 47%) characterizes the dependence of the uprooting index (*Upi*; the ratio of uprooted trees and all of the trees before the wind event) on the site conditions and topography (Table 3, Figure 5). When the *H1Tn* increases by 10 cm, the *Upi* increases by 5.4%; when the *ID* decreases by 1, the *Upi* increases 265 by 2.7% (see *β*s in Table 3).

**Table 3.** Results of the sub-optimal GAM for relative uprootings, linear part. *β*—estimate of the regression coefficient; SD(*β*) —estimate of standard deviation of *β*; *H1Tn*—topsoil thickness; ID—index of dryness.

| Variable | *β* | SD(*β*) | *t*-Statistic | *p*-Value |
|----------|------|---------|---------------|-----------|
| intercept | 0.4058 | 0.06653 | 6.0987 | <0.0001 |
| *H1Tn* | 0.0055 | 0.00259 | 2.1223 | 0.036 |
| ID | −0.0261 | 0.00998 | −2.6186 | 0.010 |

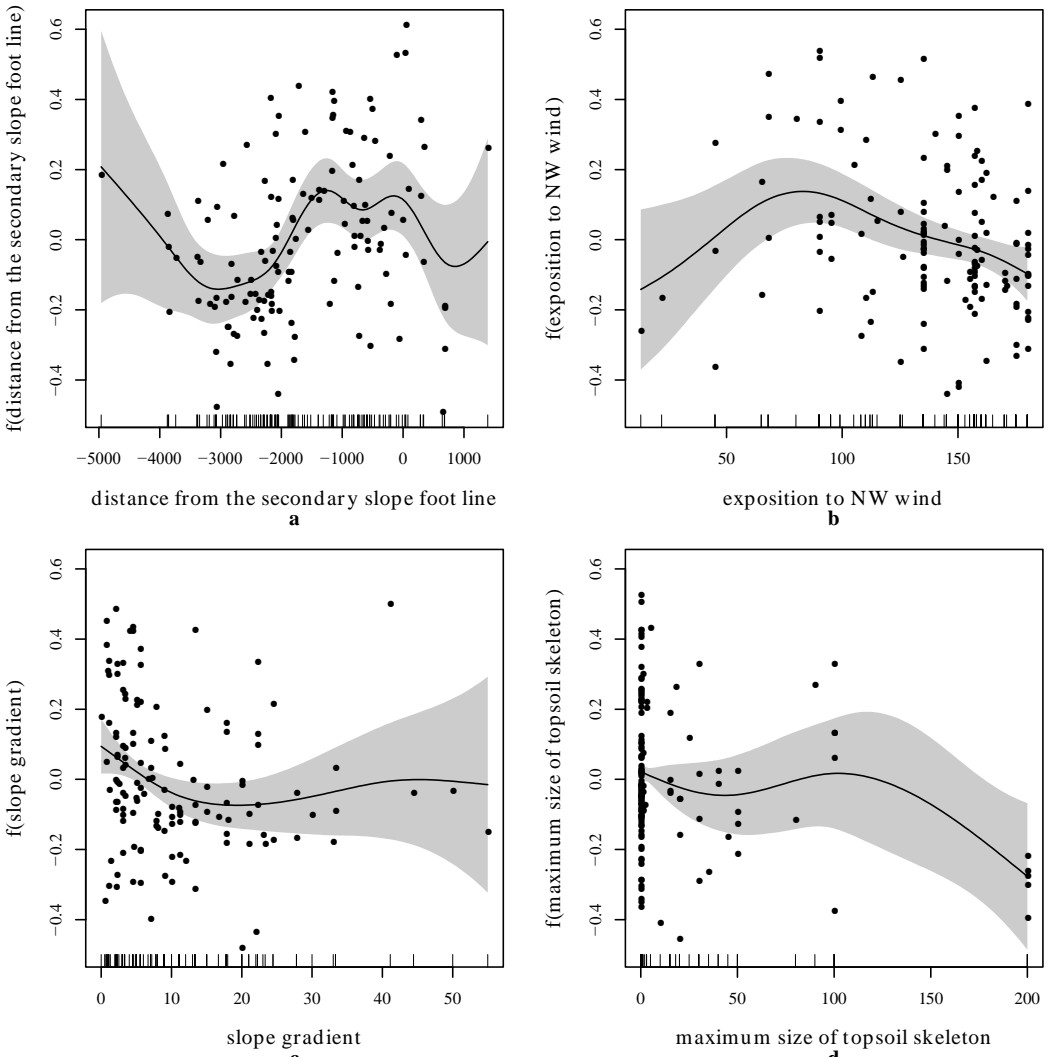

**Figure 5.** Estimated causal relationship of the relative uprootings on the particular variables (spline part of the model for *Upi*)—(**a**) distance from the secondary slope line, (**b**) exposition to NW wind, (**c**) slope gradient, (**d**) maximum size of topsoil skeleton.

In contrast with the model for *Di*, the model for *Upi* includes prevailing wind orientation (*EGNWW*), and does not contain altitude (*Alt*). The significant characteristics of the soils are moderately modified. The model contains the depth of the topsoil horizon (*H1Tn*), but does not contain the average size of the soil skeleton (Sa) or the average size of the substratum skeleton (H3Sa).

The basic interpretation of the model for *Upi* is similar to that of the model for *Di*, but more particularities exist. The interpretation of the index of dryness (ID) is the same in both models (with an increasing dryness of habitat, the number of uprooted trees also decreases), but the other soil characteristics are less important in the model for *Upi*. In contrast, the topographic characteristics are considered to be the most important characteristics in the model for *Upi*.

## 4. Discussion

The vertical gradient of the average skeleton size (*GSa*, *p*-value = 0.0032) belongs to our most important findings from the nonlinear soil effects. The damage increases with the skeleton size because a bigger skeleton in the upper soil horizons hinders the development of the root system. Less damage was recorded at the sites with smaller skeleton sizes in the topsoil compared to the substratum (see Figure 4a). The volume of the subsoil skeleton (*H2Sv*, *p*-value = 0.0024) functioned similarly,

with the fixation mechanism observable at about a 20% skeleton content, and a very strong effect visible at 60%. As a consequence of the combined effects of the root growth inhibition (less skeleton) and fixing mechanism (more skeleton), the largest number of damaged trees was recorded at sites with volumes of subsoil skeleton between 20% and 60% (Figure 4b). The damage index also decreases linearly with increasing soil dryness because of root waterlogging at wet sites. The slope gradient (G, *p*-value = 0.0027) is one of the most important independent variables. However, in this case, we may only suppose an indirect connection. Southeasterly slopes dominate throughout the territory. Therefore, the forests on more inclined slopes were generally less damaged. However, the maximum damage was recorded after gusts occurring near the flat foothills (from 0 to 5°) where the wind flows are already more or less horizontal. On the mountain side, the air masses flowed parallel to the steep slopes; therefore, the differences in damage were minimal on the slopes above 20° (Figure 4c).

The forest was mostly destroyed around the secondary slope foot line (*DSF*, *p*-value = 0.0615), that is, 2.5–4 km downward from the main slope's foot line (Figure 1). As tesserae were largely localized between both slope foot lines, the damage index increases linearly downward from the main slope foot line (*DMF*), but the relation to the secondary slope foot line is nonlinear (because of the total devastation nearly 1000 m around it, see Figure 4d). We also conclude that the last independent variable—altitude (*Alt*)—can be interpreted alike; a linear decrease of the damage index with altitude reflects a growth in altitude upward from the slope foot lines.

The distance from the secondary slope foot line (*DSF*, *p*-value = 0.0005, Figure 5a) is the most significant independent variable. The vertical flow between the slope foot lines clearly caused significantly more uprooted trees. The minimum number of uprootings occurring around the main slope foot line (*DMF*) can be explained as a consequence of the wasting effect of the bora impact, mainly creating stem breakages.

Exposure to the NW wind (ExNW, *p*-value = 0.0028) has the second most significant effect. The maximum number of uprooted trees was recorded on slopes oriented at right angles to the wind direction and their surroundings (70–110°). More stem breakages were recorded in the range of 0–70° (windward slopes). The uprooting rate gradually decreases from 110 to 180° (lee slopes, see Figure 5b). Even if the *p*-value of the slope gradient (G, *p*-value = 0.0287; Figure 5c) in the model for *Upi* is markedly higher than it is in the model for *Di*, the spline portions of both models are very similar. This signifies a similar interpretation, though it was made in the model for *Upi* with less confidence. The number of uprootings slightly increases with the increasing depth of the first soil horizon. A bigger soil skeleton can limit the root system of the trees. More uprooted trees were recorded in habitats with a larger volume of fine-grained topsoil or large stones of moraine, i.e., the maximum size of topsoil skeleton (*H1Sx*, *p*-value = 0.0198, Figure 5d). On the contrary, the rock skeleton in topsoils with a maximum size of 35 to 65 cm fixes better to the root system of spruces. The maximum amount of uprooting was recorded on the SW- and NE-oriented gentle inclining slopes, and flat sites with deep hydromorphic soils near the secondary slope foot line. The number of uprootings slightly increases with the increasing depth of the first soil horizon. A bigger soil skeleton can limit the root systems of the trees. On the contrary, a rock skeleton with a maximum size of 35 to 65 cm in the topsoil fixes better to the root system of spruces. The maximum uprooting was recorded on the SW- and NE-oriented gentle inclining slopes and flat sites with deep hydromorphic soils near the secondary slope foot line.

It may generally be concluded that the less-damaged stands were located farther from the slope foot lines on dryer and more-sloping localities with more and bigger skeletons in the topsoil, as well as deeper extremal skeleton properties (few and tiny, or many and big). Representative examples of these characteristics are the fault and erosional slope sites lying above the foot line with cambic podzols covered by natural spruce forests (especially communities of *Eu Vaccinnio-Abietion* (Oberd. 1957)). Among the most damaged areas were wet sites containing glacifluvial forms, with waterlogged spruce forests located near the secondary slope foot line.

Wind events play a key role in forest ecosystem dynamics, and are important factors for sustainable forest ecosystem management. Even if the forests in the Tatra National Park are not completely natural,

but instead partly influenced by forest management, they play an important role in the protection of the Carpathian mountain ecosystems.

From the results of Jonášová et al. [41] it is obvious that wind disturbance does not put forest continuity at risk. Recruitment densities of between several hundred and several thousand individuals per ha should be sufficient to ensure forest regeneration [53,54]. Knowledge of the long term impact of different disturbances can help to understand changes of tree abundance and understory vegetation development after the disturbance. In cleared areas with the lowest numbers of young plants, a further decrease will then indicate the need for additional planting. We can expect an increasing number of broadleaves in uncleared windthrows, which are still without vegetation and provide suitable microsites. In addition to fulfilling natural conservation purposes, national parks also have various touristic, scientific, and educational functions. For forest managers, it is relevant to describe the impact of abiotic variables in the determination of the storm's impact across the landscape, and the identification of the potential hazards that may accompany it. In several cases, the vegetation type was not useful in predicting the stand's vulnerability to wind; the most important abiotic landscape variables, such as the altitude and topographic position, proved useful for the estimation of the windthrow risk within a stand structure class, or in the absence of stand structure data [21].

The application of GAM has contributed to the development of applied statistical methods in various fields of research, e.g., [55–58]. Our characterization of the extent of the damage in relation to edaphic factors (especially the soil skeleton and water content) confirms the general findings from previous reviews. Peterson and Pickett [59] identified an increase in uprooting in wet soil conditions. Xi [60] presented similar findings. Trees growing on dry sites may have relatively bigger root systems; big skeletons create mechanical obstructions for uprootings [7]. Rottmann [61] and Konôpka et al. [62] both stated that waterlogging worsens the coherence of soil and rootsoil friction, the confluence of which drastically lowers spruce root anchorage. On the other hand, in some regions, shallow dry soils can limit root growth and cause increased tree sensibility to wind damage [62,63]. Schulte et al. [14] stated that soil factors are significant in predicting wind disturbance, but the interpretation of the topographic effects is not certain across various regions.

The results of the linear-circular correlation analyses showed the essential impact of the orientation of the georelief on damage of trees, especially on uprootings of spruces on north-western and north wind oriented stands. The specific orographic conditions and the weather situations affect the character and distribution of the damage. The exposure to the prevailing wind is related to the slope gradient, aspect, plan and profile curvature [12]. Boose et al. [64] considered the distribution topography, exposition, and wind speed as the main regional factors of wind damage. As per the review of Everham and Brokaw [7], windstorms affect both the windward and lee slopes. Since the direct impact of the slope on the amount of damage is rare, we assume an indirect effect on the degree of damage. The majority of the Tatra forests are on lee slopes. The local wind direction was illustrated by the orientation of uprooted stems (Figure 6). Additional authors, e.g., [7,65], assumed that an increase in wind damage is localized on sites near a rapid change in slope gradient; indeed, the situation in the Tatras was the same. Our results confirm the important impact of the secondary slope's foot line on uprooting.

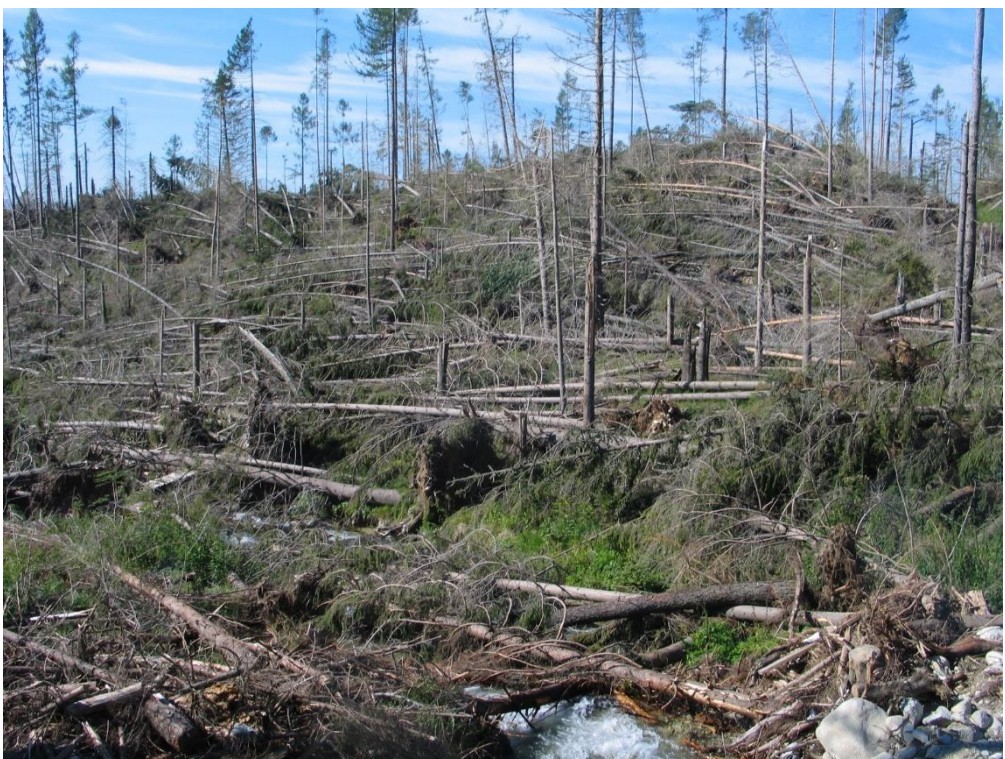

**Figure 6.** Damaged spruce forest in the Tatras foothill near Vyšné Hágy in August 2005 (photo: V. Falťan).

Although we statistically tested the effect of the values of the sine and tangent of the slope angle on the forest damage, the simple value of the slope angle proved to be most statistically significant. This shows that, in this case, the slope does not affect the distribution of the damaged forests via gravity forces. Rather, it shares the impact of the exposure to wind, but also the distance from the slope foot lines. This is indicated by the greater significance of the slope angle in model *Di*, which lacks a separate characteristic of exposure to wind, while the slope angle is less significant in the model for *Upi* (with exposure). Ulanova [4] considers that the spatial distribution of the trees was associated with the pit-and-mound topography in all types of forests after the wind disturbance. Our previously published results [42,43] confirm the protective function of massive (wet or dry) moraine depressions. This is an apparent discrepancy with our current findings (more damage was registered on wet sites); however, this is due to the fact that our wet sites were largely situated on flat waterlogged plains.

Several studies of wind disturbances are based on the analyses of aerial photos. In these cases, the damage is estimated only on a binary or ordinary scale. The ratio of damaged trees to all trees (damage index, *Di*) seems to be a much better dependent variable. It also confirms our previous results [42], where we also tested the abundance of the trees (E3) as a dependent variable, and the results were less significant.

The quality of the input database can be considered the riskiest factor of the approach presented in this research. However, despite the use of measured quantitative data along with the estimated qualitative data, the presented results are surprisingly in line with the previous results. Moreover, the interpretation of nonlinear relations carries with it a new original extension of the relations. A clear ecological interpretation of the nonlinear terms (see Section 3) is therefore essential. The GAM also enables the simultaneous fit of a spatial trend function, e.g., current flow field parameters in modeling [23]. The results of the Hurricane Katrina research [15] confirm the soil and topography (e.g., slope, aspect, physiographic position) as abiotic factors that are relevant to the control of the wind disturbance.

The evaluation of the abiotic controls on windthrow disturbance is a relevant topic for continued ecological research [8,9,13,14,22,23]. Our detailed fieldwork provides a unique dataset for research with its system approach and the usage of regression modeling techniques. Although the development of GAM is visible in biological and ecological research conducted over the last decades, e.g., [55–58], we applied GAM for the first time to the biotic and abiotic factors chosen for the models (tree height, soil depth, soil skeleton size, slope angle, and orientation) of wind disturbance in the Tatras [43]. Our further research tended towards the completion of the abiotic variables in the statistical models for this event. The maximum tree height (an affected biotic factor) was excluded from the models. We studied totally the impact of 47 independent variables on the vegetation damage, and the GAM application found 10 of them to be relevant. The extension of the models by means of other independent variables, especially the index of dryness, the average size of the soil skeleton, and the distance from the foot lines, along with the description of linear and nonlinear dependencies, significantly improved the interpretability of the results.

More detailed investigations of nonlinear dependencies between the damage and the influencing factors, the differences between uprooting and stem breakage, and the damage of various tree species are all recent challenges to wind disturbances research.

There are increasing impacts of climate change in the forests of Central Europe [66]. Climate change also leads to more frequent storms and accompanying pathogen attacks, soil erosion, and accelerated runoff. Bolte et al. [67] presented adaptive forest management, which can help managed forest ecosystems to adapt to these new conditions and reduce the risk of degradation. The results of wind event research are useful there. Forest managers can see the potential risks of windstorms on various types of sites and make decisions for managed zones of forest areas.

## 5. Conclusions

This paper dealt with the modeling of abiotic controls on wind disturbance in the Tatra National Park. From the tested models, the two most relevant controls were presented, dependent upon the relative amount of forest damage: the damage index (model for *Di*) and the uprooting index (model for *Upi*) on a set of abiotic variables. The spruces' waterlogged, shallow root system is significantly prone to uprooting. Therefore, the dryness index is statistically more important in the model for *Upi*. The skeleton affects the rate and manner of the damage (uprooting), especially on sites where the wind force had less impact. Windbreak, without relation to the substrate, prevailed in places with gale force winds. Most uprooted trees were recorded in places without a skeleton. While evaluating the abundance of the tree layer, the following was observed in the study area: there was significantly less uprooting damage recorded at the sites which had a greater abundance of *Larix decidua*. In general, exposure to the NW wind has a significant influence on uprooting and the extent of the damage to the trees during the wind disturbance.

The space in front of the secondary slope's foot line has most likely been devastated because of the turbulence resulting from the slope-break effects. The devastation generally graduates from the mountains to the gentle foothills. The use of comprehensive field data, supplemented by the set of derived attributes provided by the GAM framework, provided non-trivial knowledge. A deeper analysis of the relationships, combined with acceptable statistical robustness, is the main contribution to this research in comparison with older studies dealing with this topic. The results will help us to better understand the sensitivity of forest ecosystems to wind disturbances; to identify sites prone to damage, the subsequent soil erosion and the accelerated runoff during flash flood events; and provide knowledge for the management of the Tatra National Park.

**Author Contributions:** Conceptualization: V.F., S.K. and J.M.; methodology: V.F., J.M., S.K., and M.B.; field investigation: V.F., J.M. and M.B.; visualization: M.B., S.K. and M.M.; data analyses and interpretation: S.K., V.F., J.M., and N.P.; writing—original draft: V.F., S.K., J.M., N.P., S.Z. and F.P.; writing—review and editing: V.F., S.K., J.M. and F.P.; supervision: V.F.; project administration: V.F. and F.P. All authors have read and agreed to the published version of the manuscript.

**Funding:** This publication was funded by the Scientific Grant Agency of the Ministry of Education of the Slovak Republic and Slovak Academy of Sciences, grant VEGA 1/0247/19 "Assessment of land-use dynamics and land cover changes". The statistical analyses were funded by the Grant Agency of Masaryk University, project 384 MUNI/A/1418/2019 'Mathematical and statistical modelling 4 (MaStaMo4)'.

**Conflicts of Interest:** The authors declare no conflict of interest.

## Abbreviations

The following abbreviations are used in this manuscript. Variables used for the evaluation of abiotic controls on wind disturbance; standard writing—primary variables measured in the field; italics—variables derived from the primary variables; *r*—variables in relative scale; the orientation of the georelief = 0°—the orientation of the georelief to the north, 90°—to the east, 180°—to the south, and 270°—to the west; the orientation of the georelief equal to the direction of the north and the northwest wind = 0°—the real orientation, 180°—the opposite orientation; profile curvature—ordinary scale, levels: 1 (convex), 2 (linear) and 3 (concave); plan curvature—ordinary scale, levels: 1 (convex), 2 (linear) and 3 (concave); surface water retention—binary scale, levels: 0 (absence) and 1 (presence); index of dryness—ordinary scale, levels: from 0 (maximum moisture of habitat) to 7 (maximum dryness of habitat); (soil) texture—ordinary scale, levels: from 1 (sandy) to 7 (clay); rock skeleton weathering—ordinary scale, levels: from 0 (non-weathered rock) to 4 (completely weathered rock); rock skeleton volume—percentage of the rock skeleton in the soil horizon.

| Abbreviation | Variable Category and Name |
| --- | --- |
| | Vegetation |
| NUp | Number of uprootings |
| NSb | Number of stem breakages |
| NSt | Number of standing trees |
| TT = NUp + NSb + NSt | Total trees |
| *Di* = (NUp + NSb)/TT | Damage index [dependent variable] |
| *Upi* = NUp/TT | Uprooting index [dependent variable] |
| | Relative abundancies (to all trees) [independent variables] |
| rPicea | Relative abundance of standing European spruce (*Picea abies*) on a site |
| rPinus | Relative abundance of standing pine trees (*Pinus silvestris*) on a site |
| rLarix | Relative abundance of standing European larch (*Larix deciduas*) on a site |
| rBetula | Abundance of standing birch (*Betula pendula*) on a site |
| | Exposition (orientation) of georelief [independent variables] |
| EG (degrees, 0 to 359) | The orientation of the georelief to the north |
| EGNW (degrees, 0 to 180) | The orientation of the georelief equal to the direction of the north wind |
| *EGNWW* (degrees, 0 to 180) | The orientation of the georelief equal to the direction of the northwest wind |
| WIA (degrees, 0 to 359) | The angle of impact of the wind |
| | Topography [independent variables] |
| *Alt* (m a.s.l.) | Altitude |
| As (degrees) | Slope aspect |
| *G* (degrees) | Slope gradient |
| PrC | Profile curvature |
| PlC | Plan curvature |
| DR (m) | Distance from ridges |
| *DMF* (m) | Distance from main slope foot line |
| *DSF* (m) | Distance from secondary slope foot line |
| DV (m) | Distance from valley lines |
| | Hydrography [independent variables] |
| Wd (m) | Depth of water table |
| Sr | Surface water retention |
| *ID* = f(Wd, Sr, *G*, PrC, PlC) | Index of dryness |
| | Soil (Topsoil H1, Subsoil H2, Substratum H3) [independent variables] |
| Tx, H1Tx, H2Tx, H3Tx | Soil texture |
| STn, *H1Tn*, H2Tn, H3Tn (cm) | Soil thickness |
| Sw, H1Sw, H2Sw, H3Sw | Soil skeleton weathering |
| Sv, H1Sv, *H2Sv*, H3Sv (%) | Volume of soil skeleton |
| Sx, *H1Sx*, H2Sx, H3Sx (cm) | Maximum size of soil skeleton |
| Sa, H1Sa, H2Sa, H3Sa (cm) | Average size of soil skeleton |
| *GSv = (H1Sv − H3Sv)/STn* | *Vertical gradient of skeleton volume* |
| *GSx = (H1Sx − H3Sx)/STn* | *Vertical gradient of skeleton maximum size* |
| *GSa = (H1Sa − H3Sa)/STn* | *Vertical gradient of skeleton average size* |

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
