# Peer review of "Evaluation of Abiotic Controls on Windthrow Disturbance Using a Generalized Additive Model: A Case Study of the Tatra National Park, Slovakia"

_forests, doi:10.3390/f11121259_

Round 1

Reviewer 1 Report

This manuscript focused on estimating impacts of wind disturbance on the forests of Central Europe. Based on the detailed investigation of damaged stands, soil, and topography, the authors revealed relationships between the damage indexed uprooting index and various soil and topography properties. I think this manuscript displayed wind impact on Central Europe forests and may arouse some interests to audiences in forestry, biometrics, and ecology. It clearly falls into the scope of journal – Forests. However, some details were lacked and it needs to be revised in some places, and some parts require clarification, as outlined in the following.

  1. Please provide reasons why used a generalized additive model (GAM) from several candidate models.
  2. The source of forest inventory data was not clear, such as how many field plots were used and what summary statistics are for tree and stand-level variables.
  3. The authors only showed a vague relationship between damage indexed uprooting index and various soil and topography properties, but how to testify if wind variables (e.g. wind speed, wind direction) affected damaged stands?
  4. Please enhance the discussion on future perspective of wind impact on Central Europe forests.

Author Response

Rev 1

Authors thank the reviewer for suggestions for improving the article.

Comments and Suggestions for Authors

This manuscript focused on estimating impacts of wind disturbance on the forests of Central Europe. Based on the detailed investigation of damaged stands, soil, and topography, the authors revealed relationships between the damage indexed uprooting index and various soil and topography properties. I think this manuscript displayed wind impact on Central Europe forests and may arouse some interests to audiences in forestry, biometrics, and ecology. It clearly falls into the scope of journal – Forests. However, some details were lacked and it needs to be revised in some places, and some parts require clarification, as outlined in the following.

  1. Please provide reasons why used a generalized additive model (GAM) from several candidate models.

  1. The source of forest inventory data was not clear, such as how many field plots were used and what summary statistics are for tree and stand-level variables.

We didn’t use any summary statistics. Since the data are very skewed, using mean is far from optimal. If needed, we can add, e.g. minimum, first quartile, median or third quartile. Maximum. For directional variables, it will be more difficult since these classical summary statistics are not advised to use, but we could possibly use some directional summary statistics instead

  1. The authors only showed a vague relationship between damage indexed uprooting index and various soil and topography properties, but how to testify if wind variables (e.g. wind speed, wind direction) affected damaged stands?

The wind speed varied in time and space and cannot be directly estimated for 260 investigated sites: it can be a reason why our models are not more strong. However, we know the dominant wind direction (NW and N) and it was included into the models (variables EGNW and EGNWW in the Table 1). These variables were statistically significant.

  1. Please enhance the discussion on future perspective of wind impact on Central Europe forests.

The comment was accepted.

We enahanced the discsussion.

Reviewer 2 Report

This manuscript presents an analysis of the factors affecting forest damage in a large wind event in Tatra National Park, Slovakia. The authors applied GAMs and found strong relationships between edaphic/topographic variables and two different measures of forest damage.

While the topic is timely and relevant, I have some concerns about the current form of the manuscript. The major problem with the paper right now is organization. The large-scale structure of the paper is problematic, with large sections of the results that read more like discussion (lines 379-354). The small-scale organization is frequently problematic as well. For instance, in the introduction, the section on previous models should be condensed and ordered logically. Other instances of confusing organization are given in my line comments below. The writing has a tendency towards vagueness (e.g., line 50: "a number of hazards in different areas"), and the abbreviations are not always helpful or easy to remember (e.g., H1Tn for topsoil thickness).

I also have some concerns about the statistical methods. The analysis seems to rely heavily on Wood (2006), but the state of the art has advanced since that book was published. I have cited some more recent references in the line comments, as well as some further concerns about the analysis (see especially the comments on lines 228 and 432).

Line comments:

line 24, 108: "the bora" may not be a familiar term to readers-- consider a brief explanatory phrase, e.g., "strong northeastern katabatic winds"

line 27, 105: wind disturbance

line 34: the phrase "damage indexed uprooting index" is unclear-- is this one or two response variables?

line 37: "soil skeleton" is likely to be an unfamiliar term for many readers. It would be helpful to briefly define it.

line 40: the spruce's

line 65: cite source (or move this down to the following pargraph, where it works better as a topic sentence).

line 82: the sentence beginning here should be placed earlier in the paper.

line 95: if these data were collected in 2004, why is the analysis being performed only now? It would be useful to have some context for this decision.

lines 111-129: cite sources

line 134: fragments

line 134 passim: cite a source for these community classifications

line 146: put _Ips typographicus_ in italics

Fig. 3: shouldn't there also be a link from soil to water, since soils differ in permeability in ways that can affect water distribution in a landscape?

line 172: rock skeleton and geometric form (in this context) are unfamiliar phrases to me, and to a geologist colleague whom I asked about these phrases. It would be useful to define these variables.

line 174: exposure to the wind

line 200: what version of R?

lines 202-203, 240-244: I am confused by the use of linear-circular correlation here. It seems like WIA is the only variable that is circular. Also, it seems like there is very likely to be some collinearity between these variables since EGNW and EGNWW are probably similar to one another. With this many comparisons, it would make sense to apply a family-wise error rate correction (e.g., Benjamini-Hochberg FDR, which offers higher power than Bonferroni).

line 214: some of the approaches described by Wood (2006) have been superseded by more recent developments in the field, particularly with respect to calculation of p-values. See Wood (2013 Biometrika 100(1) 221-228) and Wood (2013 Biometrika 100(4): 1005-1010).

line 214: format citation

line 228: what criterion was used in model selection: AIC? Also, how did you account for the null space in your model selection criteria? Model selection with GAMs requires a careful approach since the null space of the spline basis is not penalized; penalties operate only on the range space (i.e., the nonlinear components of the fit). In mgcv you can use the double penalty approach (select = T) to do model selection that accounts for the null space. But you should specify this in your methods.

lines 251-254: many of these variables are not described or specified in the methods. What is, e.g., "average size of substratum skeleton," and how was it assessed?

lines 261, 310, 410: it's confusing to refer to these as model 1 and model 2, which implies that they are modeling the same thing. The first is explaining deviance in overall forest damage while the second is explaining deviance in uprooting. It would be clearer to refer to these models by their respective response variables; e.g., in line 339, "the p-value of the slope gradient in the model is markedly higher for forest damage than for uprooting, but the spline portions for both models are very similar."

lines 279-354: much of this section reads more like Discussion than Results.

line 304: "deeper extremal skeleton properties" is unclear.

line 346, 352: the result about spruces is repeated

lines 368-370: the sentence about values of national parks is an odd non sequitir here.

lines 406-407: are these results included?

lines 432-436: I'm not sure it's statistically legitimate to compare adjusted R2 between GAMs and OLS models. One issue is that GAMs are typically not penalized for parameters in the null space (i.e., the linear parts of the fit), and so they can add more parameters overall-- thus leading to a better fit.

line 456: use Latin names

line 456: exposure to the NW wind

Author Response

Rev 2

Authors thank the reviewer for suggestions for improving the article.

This manuscript presents an analysis of the factors affecting forest damage in a large wind event in Tatra National Park, Slovakia. The authors applied GAMs and found strong relationships between edaphic/topographic variables and two different measures of forest damage.

While the topic is timely and relevant, I have some concerns about the current form of the manuscript. The major problem with the paper right now is organization. The large-scale structure of the paper is problematic, with large sections of the results that read more like discussion (lines 379-354). The small-scale organization is frequently problematic as well. For instance, in the introduction, the section on previous models should be condensed and ordered logically. Other instances of confusing organization are given in my line comments below. The writing has a tendency towards vagueness (e.g., line 50: "a number of hazards in different areas"), and the abbreviations are not always helpful or easy to remember (e.g., H1Tn for topsoil thickness).

The organisation of the paper was changed and English sentences in the paper were corrected..

I also have some concerns about the statistical methods. The analysis seems to rely heavily on Wood (2006), but the state of the art has advanced since that book was published. I have cited some more recent references in the line comments, as well as some further concerns about the analysis (see especially the comments on lines 228 and 432).

See also comments below about null space and p-values. We actually used Wood 2013 and model selection was corrected based on Marra and Wood 2013 Computational Statistics & Data Analysis, 55(7), 2372–238. We modified the model selection base on your suggestion below.

Line comments:

line 24, 108: "the bora" may not be a familiar term to readers-- consider a brief explanatory phrase, e.g., "strong northeastern katabatic winds"

The comment was accepted.

line 27, 105: wind disturbance

The comment was accepted.

line 34: the phrase "damage indexed uprooting index" is unclear-- is this one or two response variables?

There are two response variables, the part of sentece was corrected „damage index and uprooting index

line 37: "soil skeleton" is likely to be an unfamiliar term for many readers. It would be helpful to briefly define it.

The comment was accepted.

Soil skeleton are rock fragments contained in the soil. It is added into the Abstract

line 65: cite source (or move this down to the following pargraph, where it works better as a topic sentence).

The sentence was moved down to the following paragraph.

line 82: the sentence beginning here should be placed earlier in the paper.

The sentence was placed earlier.

line 95: if these data were collected in 2004, why is the analysis being performed only now? It would be useful to have some context for this decision. Moderné metódy GAM a niekoľkoročný detailný terénny výskum spolu s realizáciou výskumných tesserae260 

Data were collected during several years of detailed field research on the 260 research points.

lines 111-129: cite sources

The comment was accepted

line 134: fragments

The comment was accepted

line 134 passim: cite a source for these community classifications

Source is: Mucina, L., Maglocký, Š, eds. (1985):A list of vegetation units of Slovakia. Doc. Phytosociol., Camerino, 9: 175-220.

line 146: put _Ips typographicus_ in italics

The comment was accepted

Fig. 3: shouldn't there also be a link from soil to water, since soils differ in permeability in ways that can affect water distribution in a landscape?

Yes, it is relevant suggesion: it was added to the Fig. 3.

line 172: rock skeleton and geometric form (in this context) are unfamiliar phrases to me, and to a geologist colleague whom I asked about these phrases. It would be useful to define these variables

The comment was added:

line 174: exposure to the wind

The comment was accepted

line 200: what version of R?

lines 202-203, 240-244: I am confused by the use of linear-circular correlation here. It seems like WIA is the only variable that is circular. Also, it seems like there is very likely to be some collinearity between these variables since EGNW and EGNWW are probably similar to one another. With this many comparisons, it would make sense to apply a family-wise error rate correction (e.g., Benjamini-Hochberg FDR, which offers higher power than Bonferroni).

Yes, p-values might be corrected for multiplicity. Usually, the reader could perform the correction based on the preferred method. The simplest way would be to use Bonferroni method since we calculated the correlation coefficients for four variables seven times, we amended the table with a sentence „The significance level was Bonferroni corrected by dividing 0.05 by four (the table is divided to seven sections by four correlations), which resulted in 0.0125. All p-values smaller than this level indicate statistically significant correlations at a Bonferroni corrected significance level of 0.0125.“

line 214: some of the approaches described by Wood (2006) have been superseded by more recent developments in the field, particularly with respect to calculation of p-values. See Wood (2013 Biometrika 100(1) 221-228) and Wood (2013 Biometrika 100(4): 1005-1010).

Thank you for interesting suggestion. The first paper is relevant but not cited by Simon Wood in newest version of mgcv package (see R help) but in Marra and Wood 2013 Computational Statistics & Data Analysis, 55(7), 2372–238. The double penalty approach and p-value calculation is implemented in mgcv based on this paper. The second paper is interesting but not relevant since it is about random effects which we don’t have. We have only fixed effects. We actually used the second edition of Simon Wood bool from 2017. We fixed it in the paper and also changed ref. [50] to Marra and Wood 2013 mentioned above.

line 214: format citation

The comment was accepted

line 228: what criterion was used in model selection: AIC? Also, how did you account for the null space in your model selection criteria? Model selection with GAMs requires a careful approach since the null space of the spline basis is not penalized; penalties operate only on the range space (i.e., the nonlinear components of the fit). In mgcv you can use the double penalty approach (select = T) to do model selection that accounts for the null space. But you should specify this in your methods.

Thank you for pointing this out. This is indeed very important. We used AIC. We recalculated model selection and final p-values. In model for Upi it didn’t change the suboptimal model only p-values.

lines 251-254: many of these variables are not described or specified in the methods. What is, e.g., "average size of substratum skeleton," and how was it assessed?

the average skeleton size was calculated as the average of the measured skeleton sizes (in soil horizons – topsoil, subsoil, substratum), the description of the measurement was specified in Variables.

lines 261, 310, 410: it's confusing to refer to these as model 1 and model 2, which implies that they are modeling the same thing. The first is explaining deviance in overall forest damage while the second is explaining deviance in uprooting. It would be clearer to refer to these models by their respective response variables; e.g., in line 339, "the p-value of the slope gradient in the model is markedly higher for forest damage than for uprooting, but the spline portions for both models are very similar."

We incorporated different notation for the models, Model 1 is model for Di and Model 2 is now model for Upi.

lines 279-354: much of this section reads more like Discussion than Results.

The relevant parts close discussion of this section were moved into DIscussion.

line 304: "deeper extremal skeleton properties" is unclear.

It is explained in parentheses.

line 346, 352: the result about spruces is repeated

The comment was accepted

lines 368-370: the sentence about values of national parks is an odd non sequitir here.

The comment was accepted

lines 406-407: are these results included?

Distance from the secondary is confirmed as relevant in both models.

lines 432-436: I'm not sure it's statistically legitimate to compare adjusted R2 between GAMs and OLS models. One issue is that GAMs are typically not penalized for parameters in the null space (i.e., the linear parts of the fit), and so they can add more parameters overall-- thus leading to a better fit.

We deleted the sentence about this comparison. We corrected the p/values with respect to null space as mentioned above.

line 456: use Latin names

The comment was accepted

line 456: exposure to the NW wind

The comment was accepted

Reviewer 3 Report

Comments attached

Author Response

Authors thank the reviewer for suggestions for improving the article.

Evaluation of abiotic controls on windthrow disturbance by generalized additive model 

(A case study in the Tatra National Park, Slovakia) 

This study is a valuable contribution to knowledge of abiotic powers affecting forest  vegetation. While local (the case study), it can be extended regionally (Central European) and  even globally. With the use of GAM, this work is novel.   However, the study suffers from formal and also informal drawbacks. Major revision is  needed. 

General comments 

Terminology 

There is a quite number of terms in the manuscript such as ecosystem, geosystem  geoecology, biogeography. I get a difference between eco and geosystem, but the use of the  term geoecology sounds nor clear for me, especially in connection with a term biogeography. 

We simplified the terminology and abandoned terms geoecology and geosystem.

In L 89-90 you explain connection of the approaches as Geoecological conditions (including  soil, topography, and climate)My suggestion: stay simple with the terminology and avoid heterogeneity of it. Use clean terms to characterize and describe you study. 

Most of the authors come from the geographical community, but due to the comments we have reflected them for better readability of the article.

Grammar 

While reading of English is smooth, there are some grammar mistakes especially missing and misused articles. Please revise. I did not check the References section. 

English language in the paper was corrected.

Title 

I suggest omission of a part of the title to the form: “Evaluation of abiotic controls on windthrow disturbance: a case study in the Tatra National Park, Slovakia” 

New title is

„Evaluation of abiotic controls on windthrow disturbance using a generalized additive model: a case study of the Tatra National Park, Slovakia“

The application of geenralized additive model has brought significant improvements for models in the Tatras, so we would like to keep this term in the title.

Abstract 

Please consider adequate changes corresponding to comments in the next text. Mainly, try to reduce the abstract body. I miss here some essence of the article such as the  benefit/value of the study. 

 The comment was accepted.

  1. Introduction

You should clearly state the research goals instead what you did (L 97-98). This sounds like  results. Please see the specific comments.

The comment was accepted.

  1. Material and Methods

2.1. Wind event and study area.  

Make the subchapter more consistent. E.g., in L 103-106 you talk about vegetation. The same 

for L 130-140.The storm story is interesting but please consider reduction of this story, I think it is too long and Fig. 2 is not necessary. I would appreciate more information on geology and soils at the expense of the storm story. 

Your study is built on geoecology, right? Two bare sentences are not enough even with the reference.  

2.2. Variables. You may consider to rename it for Data  

Sub-chapter name „Variabples“correspond with following sub-chapter „Models“.

Hydrological and soil properties is a very important part of the research. I miss the information how did you get them by measurements and estimation (L 170-173 and the  abbreviation section). Did you dig a soil pit at your tesserae? Or? Did you analyze the  samples in a lab? 

A part of the section sounds like Results. Please move out. 

How did you transform variables? 

 We dug the soil pits and analysed only in the field.

2.3. Models 

This is very important part of the paper. In this form, it is difficult to follow (L 202-213). You  need to introduce explicitly and clearly the Models; how did you built them? You should   simplify the description of the GAM procedure. Additionally, please state explicitly what  software you used in the analysis. How did you deal with variables within the statistical  procedure? 

Please see the specific comments. 

 The software is specified in the first line of the section 2.3. Models with added details as requested by other reviewer (R version plus we added also info about Rstudio). Also used R package was specified down below the same section. The section was extended based on a request of the other reviewer towards describing model selection, including criteria used, p-values calculation etc. Now the section is clearer and easier to read.

  1. Results

The chapters 3.2. and 3.3. have the same title. Consider renaming, e.g., Model1, Model2. Some parts seems like Discussion rather than Results. Please state results supported by  numbers and then explain and discuss in the Discussion. 

Please see the specific comments. 

 The comment was accepted.

Discussion 

This sections needs a rearrangement to make it homogeneous.Additionally, whereas it is nice to reveal possibilities of a further research in a way of nonlinear relations, I miss a reason why. Why and what for is this important? What can managers and decision makers do? Can they protect forests against such a catastrophic disturbance? You can put a word about natural powers, which can’t be controlled by any management (you have mentioned the hurricane Katrina). Are those disturbance events  (Katrina and Tatra) similar? 

Please see the specific comments. 

Discussion was rearranged . Please, see reactions to the specific comment. Managers can see potential risk of windstorms on vartious types of sites and make decisions for manged zones of the national park.  

Conclusion 

This section is kind of long and detailed. Please reduce for strong statements only. 

 The comment was accepted.

Abbreviations 

There are some digits within the text before the table, maybe as a residuum of something. Consider this as a regular Table in the text. Then, you don’t have to put a legend to Tables 2  and 3. Please check if you use all the abbreviation, e.g., Sw, GSx, GSv, Gsa. 

The digits are only in front of table caption indicating lines (left side margin). We can’t see other digits. The description of the variables from table 1 was deleted, all abbreviations are explained in the Abbreviation section as was suggested in LaTeX template on the journal web. Related to the abbreviation mentioned above – Gsa doesn’t exist but GSa does and is used in text and mentioned in the Abbreviation table; Sw, GSx, GSv are in Abbreviation table and were used in the saturated model from which we created sub optimal model in model selection process. We added a sentence about model selection in the section 2.3. Models.

Specific comments 

L21-28. Please reduce this section to a couple of sentences.  

Section was reduced

L27-28. This sentence makes no sense here in the Abstract. 

The sentence has been rewritten.

L42-43. Check for typos. 

Typos were checked

L46. Disturbances are a part of the ecosystem concept. I would replace the word ‘ecosystem’ 

for ‘vegetation’. 

Word was replaced

L55. Stand structure …what about….a composition?  

 Words were replaced

L58. Tree species. 

The comment was accepted.

L60. Wind velocity and…direction? 

The comment was accepted.

L66. Please and citation names for [10]. The same for next similar cases. 

The comment was accepted.

L71.  …in other types of forest stands …such as?  

 The comment was accepted.

L78. Typo. 

The comment was accepted.

L90. The expression “…functional biogeography of plants and ecosystems” is not fortunate. I 

can imagine functional biogeography of ecosystems, but plants? See plant functional types, 

traits etc.  

The comment was accepted.

L92-95. This part is methodological, please remove. 

The part was removed.

L95. No parentheses in Slovakia . 

The parentheses were removed.

L95-98. This part sounds like results. Reword it for goals. 

The part was partially removed and rewritten.

L103-106. Why was spruce monocultures planted in the outskirts of NP? The same age and 

density are not the very factor causing spruce monocultures’ ecological lability.  

Spruce monocultures were planted in meadows and cuttings after a wind disaster at the beginning of the last century. It means the lability of a monoculture in relation to a strong wind. The trees are thicker next to each other and usually also have a slightly higher center of gravity, which causes a greater risk of damage by strong winds..

L117. Is the Figure 2 necessary to post?  

The synoptic situation from the Slovak Hydrometeorological Institute confirms the wind directions used as independent variables in the models and illustrates the wind event in a broad context

L118. Is it possible to face to such a strong wind at all?  Even in close-to-natural forests? 

The diverse structure with gaps between tree individuals along with their position relative to the windstorm can to some extent mitigate the effects of the calamity.

L132. Is ‘undisturbed’ a right word? It was disturbed already.…Consider change for e.g.,  unmanaged, left alone. 

 The comment was accepted.

L136-137. In 2004. This is redundant. We know that already. 

 The comment was accepted.

L141-142. This bare information is not enough. 

More relevant information is stated about wind event

 L142-144. Meaning of this sentence? Are these your areas, parts of your research? If so  please remove this sentence to a proper part of the section. If not, is it important to keep? 

These areas were located due to the great scientific interest in long-term research in the area affected by the wind disaster.

L146. The scientific name in italic. Check further for the same issue. 

 The comment was accepted.

L147. What is special geographic form of spruce forests? A specific community, perhaps? 

 The comment was accepted.

L 150.What is difference between windthrow and tree extraction? 

 windthrow represents wind damage, tree extraction represents logging

Figure 1. The legend - what is ‘main/secondary slope foot line. It is important part of the study but never introduced. How did you derived the lines? 

The Figure 1 with descritption was corrected.

L157. I have never seen a term ‘tesserae’. Sample plot? You call that so in L164. 

 The term tesserae was abandoned.

L158-159. How did you localize the tesserae? See L 164. Damaged geosystems. I think ecosystems were damaged, weren’t they? 

The comment was accepted and localization was described

Figure 2 . Is it necessary to post? 

The synoptic situation from the Slovak Hydrometeorological Institute confirms the wind directions used as independent variables in the models and illustrates the wind event in a broad context more than words

L168-169. Was the selection of these types of variables supervised by something? Literature, own experience? It would be good to cite sources of you decision or even your own 

experience. Not clear to me now. 

Both on literature and our study

L171-173 How did you get soil variables? Digging a soil pit? Did you analyzed soil samples  at the place in the field? Or did you use a soil lab for determination and analysis of your soil  samples? Please clarify that including the methods used. What is geometric form?  

 All information added, one soil pit per sample plot, soil variables determined in the field. No soil lab was used for analysis of soil samples. Methods are clarified on lines 171 – 186.

L174. How were the secondary variables derived? 

secondary variables were subsequently derived from the primary variables, e.g. exposure to the wind, index of dryness, etc. (see Abbreviations), L213-214.

L176-177. Dtto as Figure 1. How were the lines distinguished?  

The Figure 1 with description was corrected. The primary foot line follow tectonic boundary between Podtatranská kotlina (basin) and Tatras. The secondary foot line follow front of Tatra’s glacial moraines.

L 187 We estimated/measured also abundance of major/dominant tree species…. 

The sentece was rewritten

L 187-191. This part is a clear result. Move. 

The part was moved into results

L191-197. This part is hanging in the air somehow. If the wind direction is so important,  move that part to a proper place like L161 or so. 

 The part was moved  to proper place

L198.  ‘All other’ variables are not described in the Abbreviation section. Actually, there are  ALL and other variables there, even those you did not used. 

 Thank you for pointing this out. As mentioned above, all variables in the section Abbreviation were used. Some in calculation of derived variables, and all in models selection process. This is now indicated in the section 2.3. Models.

L201. Is this [38] the right citation? Should not be [47]? 

 The comment was accepted.

L203. The association was assessed instead measured? What is a linear-circular correlation?  A word of it would be nice. What is its advantage against conventional cor. coefficients? 

 The term “measured” was changed to “assessed”. The term linear-circular correlation is basic term in directional statistics and is assessing correlation between linear and directional (circular) variable. We added this sentence to the section 2.3. “In this situation classical (linear-linear) correlation coefficient can’t be used, since the variables related to the orientation are directional variables.”.

L205. All alternative hypotheses…what are they? Where are they stated? 

 The alternative hypotheses are always the opposite of the null and stated in the paper in the section 2.3. as “The null hypothesis that the correlation coefficient is equal to zero against two-sided alternative”. Similarly, we defined state for GAM parameters “The null hypotheses were tested against two-sided alternatives”. We think that it is clear statement about hypotheses but to add clarity about hypotheses related to the smooth functions we also added “The null hypotheses for smooth terms are defined such that it is not a (potentially non-linear) association of dependent variable with a covariate (independent variable), given (potentially non-linear) association with other covariates.”.

L207.  Separated to quadrants…this is a new thing. What are these quadrants? 

Quadrants are word from geography for orientation of georelief (or wind quadrants eg. NE, SE, SW, NW).

L209.  ‘Class labels’. Again, a new term, not introduced before…? 

It is term from statistical analyses

L212-213. This sentence is a result. 

This is description – definition of model building.

L2 14. Format of the citation. 

 The comment was accepted.

L220-232. This GAM explanation is difficult to follow. Can you somehow simplify that? 

 We modified section 2.3 based on the suggestions of the two reviewers and added clarity to the process of selection of variables, p-values and hypotheses. Now the section is more readable and the flow of the process of the analyses seems to us as clear.

L221. It relaxes…a proper term? 

 Yes, it is proper term used in this context in statistics, where “relax” mean “loosen” or “free”.

L224. The individual additive terms…as penalized least squares…what is it in particular?  Variables? Is it the same thing as in L 227? 

 The term “additive terms” is a proper and often used term used in statistics referring to all covariates (independent variables) pre-multiplied with regression coefficients and smooth functions of covariates added one after another to the model. In the sentence we meant only and smooth functions of covariates since the rest of this sentence is related to the smoothing. We added “all smooth functions of covariates” to the sentence.

L229-231. Non-significant terms…how can be non-significant when they improve the model  quality? Please explain. 

 This is basic concept in statistical modelling, where any covariate included in the model improve the model in a way that the error is reduced proportionally to the influence of the added covariate. It doesn’t matter if the covariate is statistically significant or not. Often, marginally significant covariates (p-value between 0.05 and 0.1, or often also up to 0.2 or a bit higher) are included in the model since the model selecting procedures are taking them as “important”.

L232. Why did you choose the level of 0.1 and not 0.05? 

 We modified the significance statement in a way it is more traditional. We added a sentencec “… at significance level equal to 0.05, p-values were calculated as in [52], section 2.5. We consider p-values between 0.05 and 0.1 as marginally statistically significant, other higher p-values in the model as statistically non-significant but having valuable contribution to the model quality. “

L237-238. This sentence belongs to the Discussion. 

 The comment was accepted

L240-244. Table 1. Rearrangement of the title is needed. The order of abbreviation should be  consistent. 

 Since all abbreviation are in the section Abbreviation, we deleted these from the table.

L256. Typo. 

Typo was fixed

L258-259. How do you know that? Do you have any statistics for that claim?  

The sentence was removed

L264. Table 2. Why estimate of the regression coefficient. You calculated it so, I would call  that assessment or so. 

 In statistics, the regression coefficients or other parameters are always estimated using some algorithm base on likelihood function or other methods. There is no need to change classical statistical terminology here.

L271. This is Discussion. 

L279-280. Discussion. 

L302-308. Discussion. 

For all discussion comments:  parts of results closes to disscusion were moved 

L317-320. This paragraph needs to be shifted to the Methods. 

This is a part suitable for Results. Methods describe used statistical methods, Results describe real content of  models and itheir values.

L327. Typo. 

 The comment was accepted

L321-325. Consider removal to the end of the chapter. 

 The results were reorganized.

L327-330. Discussion. 

 parts of results closes to disscusion were moved 

L372. What do you mean by ‘forest type’? Actual vegetation or site conditions? 

 The comment was accepted. Vegetation

L376-381. This is a technical part about GAM. Consider link with a similar part in L 427-

  1.  

Results were shortened

L383. Why did you exclude maximum height from the models? 

 because it was the maximum height of trees on the site determined only after the calamity

L397. What is ‘plan’? 

‘plan curvature‘ is planform curvature (geomofologic.)

  1. What is distribution topography?

Distribution of local topographical condition

L399-400. Windstorms affect winward and lee slopes. Isn’t it opposite? 

Based on the results from review Everham & Brokaw, windstorms can affect both windward and lee slopes. In case of Tatras lee slopes of National park also were affected.

L406-407. This sentence is not clear. Reword, explain. 

 The comment was accepted

L433-434. This sounds like a result. 

 More information is in the results, it's just a link. 

Round 2

Reviewer 1 Report

After the revision, I think the authors have addressed most of my comments. While I still concern the introduction and discussion parts. In the introduction, the authors don’t point out if the windthrow studies are limited or rare and why this work is important and is different with existing studies. Regarding the discussion part, it seems the authors only repeated several results and listed previous findings, while there is no discussion for creativity and also no highlight why this study is more advanced than prior researches. Besides, please remove Sa average size of soil skeleton below the equation 1.

Author Response

Authors thank the reviewer for another suggestions for improving the article.

After the revision, I think the authors have addressed most of my comments. While I still concern the introduction and discussion parts.

In the introduction, the authors don’t point out if the windthrow studies are limited or rare and why this work is important and is different with existing studies.

Thanks for the comment. We added requested information – L 92 – 99.

Previous research has mostly focused on the analytical investigation of chosen biotic or abiotic factors. In order to bring qualitatively new complex results, in our research we focused on combining several methodological approaches with the task of capturing the influence of topographic and edaphic factors on the degree of storm damage to forest vegetation more comprehensively. We suppose, that complex large-scaled field research, usage of geographical information systems  and actual  statistical methods can help to describe both linear and nonlinear relationships in detail.

Regarding the discussion part, it seems the authors only repeated several results and listed previous findings, while there is no discussion for creativity and also no highlight why this study is more advanced than prior researches.

Thanks for the comment. For the previous recommendations of other reviewers, only the statements in the results remained, our comments on the results were moved to the beginning of the discussion. In the next part we confront the findings with the works of other authors. We slightly rearranged discussion and at the end we added another highlights why this study is more advanced than prior research (especially, that we studied totally impact of 47 independent variables on vegetation damage and GAM application found together 10 relevant of them. The extension of the models by means of other independent variables, especially the index of dryness, the average size of the soil skeleton, and distance from the foot lines along with description linear and nonlinear dependencies significantly improved the interpretability of the results.).

Besides, please remove Sa average size of soil skeleton below the equation 1.

Thank you, the missing variable name was deleted.

Reviewer 3 Report

Dear Authors,

Thanks for the comments and work on the paper improvement.

I still have some minor comments.

Please see the attached document with the comment boxes.

Regards

Author Response

Authors thank the reviewer for another suggestions for improving the article.

Evaluation of abiotic controls on windthrow disturbance by generalized additive model (A case study in the Tatra National Park, Slovakia)

Abstract

You should mention the essence/benefit/added value of the study there. I cannot see any adition of that.

Thanks for comment. We added it to Abstract

Introduction

I am not happy with the formulation of the goals. The last paragraph is methodological. Doesn’t belong here.

The last paragraph was moved to Methods. The paragraph was added as a request of another reviewer.

Material and Methods

2.1. Wind event and study area

Fig. 1. Title, please make sure, it is clear.

Thank you for comment. We clarified title and legend of Figúre 1

Make the subchapter more consistent. E.g., in L 103-106 you talk about vegetation. The same for L 130-140.The storm story is interesting but please consider reduction of this story, I think it is too long and Fig. 2 is not necessary. I would appreciate more information on geology and soils at the expense of the storm story. Your study is built on geoecology, right? Two bare sentences are not enough even with the reference. Sorry. My comments were not accepted.

Thank you for comment. Sorry, we forgot this earlier. The text about vegetation and storm has been shortened. Because of the shortened text, the Fig 2 is, in our view, necessary for a better understanding  of the metherological situation. We have added information about geology and soil and their impact on spatial extent of vegetation, land cover changes after windstorm from specialized paper [35] from authors which are also members of our research team.

2.2. Variables

How did you transform variables? Did you adress this?

There was no need to transform the variables in any way. We used the original scale of measured data and also calculated data. Since we didn’t perform any transformation, it is not mentioned in Methods.

Results

Table 1

The title is not in a good shape. It should be presentable on its own, a note where one can find the variable abbreviations is necessary.

Thanks for the comment. We put back the explanation of the abbreviation and make sure they are in the right order (as you requested in previous comments).

Additionally, L 288-289, based on what (the tree abundance?). you can claim that the max. height of standing trees was the most important factor. Can I find it in Table 1? Can you explain the logic of that statement?

In our previous researches, we evaluated also the max height of standing trees and its impact to number of standing trees after the storm. But it was the maximum height of trees on the site determined only after the calamity. It was not fortunate solution as we did not have data on the current height of the trees before storm. Maximum tree height after storm (an affected biotic factor) was not included in the models (please, see discussion L 485-497). Our actual manuscript studies impact of the abiotic factors on damage of vegetation and max height of tree is not included.

Discussion

I am not sure if the shift of large parts of the Results to the Discussion was fortunate. I have suggested only some discussion-close sentences.

For the previous recommendations of other reviewer, only the statements in the results remained, our comments on the results were moved to the beginning of the discussion. We slightly rearranged discussion and at the end we added another highlights why this study is more advanced than prior research (especially, that we studied totally impact of 47 independent variables on vegetation damage and GAM application found together 10 relevant of them. The extension of the models by means of other independent variables, especially the index of dryness, the average size of the soil skeleton, and distance from the foot lines along with description linear and nonlinear dependencies significantly improved the interpretability of the results.).

L176-177. Dtto as Figure 1. How were the lines distinguished?

The footlines represent boundaries between differently inclined aeras, usually at the contact of a hillside and a valley. The primary foot line follow tectonic boundary between Podtatranská kotlina (basin) and Tatras. The secondary foot line follow front of Tatra’s glacial moraines.

Red lines ? Full red line? Where are othe ones?

The Figure 1 description and text  was corrected.
